# Heavy quasiparticles and cascades without symmetry breaking in twisted bilayer graphene

Anushree Datta[1,2,3], M. J. Calderón [1], A. Camjayi[4,5] & E. Bascones [1]✉

Among the variety of correlated states exhibited by twisted bilayer graphene, cascades in the spectroscopic properties and in the electronic compressibility occur over larger ranges of energy, twist angle and temperature compared to other effects. This suggests a hierarchy of phenomena. Using a combined dynamical mean-field theory and Hartree calculation, we show that the spectral weight reorganisation associated with the formation of local moments and heavy quasiparticles can explain the cascade of electronic resets without invoking symmetry breaking orders. The phenomena reproduced here include the cascade flow of spectral weight, the oscillations of remote band energies, and the asymmetric jumps of the inverse compressibility. We also predict a strong momentum differentiation in the incoherent spectral weight associated with the fragile topology of twisted bilayer graphene.

A large variety of correlated states including insulating, super-conducting and topological phases[1–14] have been detected in twisted bilayer graphene (TBG). Among these phenomena, scanning tunneling microscopy (STM) experiments find a strong doping dependent broadening of the density of states (DOS) with a spectral weight reorganization up to several tens of meV[6–8,15]. This reorganization happens in the form of cascades of spectral weight with resets at integer fillings[10,11], band flattening[16] and minima of the DOS at the Fermi level $DOS(\omega = 0)$ at integer dopings[8,10], and it is accompanied by oscillations in the energies of the remote bands[10]. At the charge neutrality point (CNP) the spectral weight is pushed away from low energies[10]. The cascades are revealed in scanning electron transistor (SET) experiments as strong asymmetric jumps of the local electronic compressibility[17–19].

The cascades are detected in a wider range of angles and up to much higher temperatures than the ones where superconductivity, correlated insulators and anomalous Hall effects occur[17,18,20], suggesting that the cascades constitute the parent state in which these low temperature phenomena emerge[10,17]. The cascades have been primarily interpreted in terms of symmetry breaking states[6,8,11,16,17,21–23] but also as a consequence of strong correlations in the normal

state[7,10]. The discussion on whether the phenomenology of TBG can be explained just in terms of ordered states or requires taking into account the strong modification of the normal state, put forward early on[1,7,8,24,25], is still unsettled. Progress has been hindered by the complexity of the minimal models in TBG and of the techniques required to address strong correlations beyond the symmetry breaking transitions.

The heavy fermion description for TBG, early proposed in[25] and supported by an analysis of the interactions[26], is now receiving more attention, particularly after the publication of ref. 27. In the different heavy fermion models proposed for TBG[25–30], the flat bands of each valley are dominated by two $p_+$ and $p_-$ orbitals centered at the AA regions of the moiré unit cell, except close to Γ, see Supplementary Fig. 1b. These $AA_p$ orbitals, strongly sensitive to electronic interactions, are hybridized to other less correlated electrons. In such a system, a strong spectral weight reorganization is expected even in the non-ordered *normal* state[31]. To address theoretically the relevance of these effects, it is convenient to suppress the symmetry breaking processes, either by hand in the calculation or working at temperatures above the ordering transitions. A technique particularly suitable to describe the modification of the energy spectrum driven by the local correlations,

[1]Instituto de Ciencia de Materiales de Madrid (ICMM). Consejo Superior de Investigaciones Científicas (CSIC), Sor Juana Inés de la Cruz 3, 28049 Madrid, Spain. [2]Université Paris-Cité, CNRS, Laboratoire Matériaux et Phénomenes Quantiques, 75013 Paris, France. [3]Université Paris-Saclay, CNRS, Laboratoire de Physique des Solides, 91405 Orsay, France. [4]Universidad de Buenos Aires, Ciclo Básico Común, Buenos Aires, Argentina. [5]CONICET - Universidad de Buenos Aires, Instituto de Física de Buenos Aires (IFIBA), Buenos Aires, Argentina. ✉e-mail: leni.bascones@csic.es

able to capture the frequency dependence of the self-energy, is dynamical mean-field theory (DMFT)[25,32–34]. In TBG, besides the intra unit cell interaction $U$ among the $AA_p$ orbitals, responsible for the unconventional behavior, there are other sizable interactions which cannot be neglected[26], rendering a complete DMFT treatment impossible.

Here we use a DMFT+Hartree approximation to study a multi-orbital model for TBG[26,29]. The intra and inter-orbital intra-unit cell interactions $U$ between the $AA_p$ orbitals are treated with DMFT and the other considered interactions are included at the Hartree level. Self-consistency is enforced within the DMFT loop and between the DMFT and the Hartree schemes. We fit the low energy bandstructure of TBG with twist angle $\theta = 1.08°$[35–37], Fig. 1a, with an 8 orbital model per valley and spin, adapted from[29,38]. Including the valley degeneracy, for each spin there are 4 strongly correlated $AA_p$ orbitals, degenerate in the absence of inversion or time-reversal symmetry breaking, and 12 less correlated orbitals, named $lc$ orbitals in the following. The interactions are calculated assuming a $1/r$ interaction between the electrons in the carbon atoms and a dielectric constant $\epsilon = 12$[26]. This gives a $U = 44.5$ meV, which is larger than the gap between the flat and remote bands $\Delta = 22$ meV. Calculations are done at $T = 6$ K, except otherwise stated,

and we do not allow states with spontaneous symmetry breaking. See Methods and Supplementary Fig. 1 for further details.

## Results

The DOS in Fig. 1b displays a strong energy $\omega$ and doping $v$ dependence with resets of spectral weight and minima at the Fermi level, $\omega = 0$, at integer dopings, Fig. 2a. In spite of the narrow bandwidth of the flat bands in the non-interacting model, 1 meV at M and 8 meV at $\Gamma$, Fig. 1a and Supplementary Fig. 1c, an important reorganization of the spectral weight is visible in Fig. 1b within a range of 50 meV around the Fermi level. This spectral weight appears in the form of cascades at positive and negative energies flowing from $\omega \approx \pm U$ (red arrows in Fig. 1) towards $\omega = 0$. For hole doping the spectral weight in the cascade at positive energies is larger than at negative energies and it increases with doping. This positive energy cascade forms at energy $U$ around a given integer doping and gets very close to $\omega = 0$ at the next smaller integer doping. The negative energy cascade is shifted in doping with respect to the one at positive energies and reaches $\omega = 0$ at intermediate fillings. The $v$ and $\omega$ dependence of the cascades is reversed for electron-hole doping. At the CNP ($v = 0$) the cascades

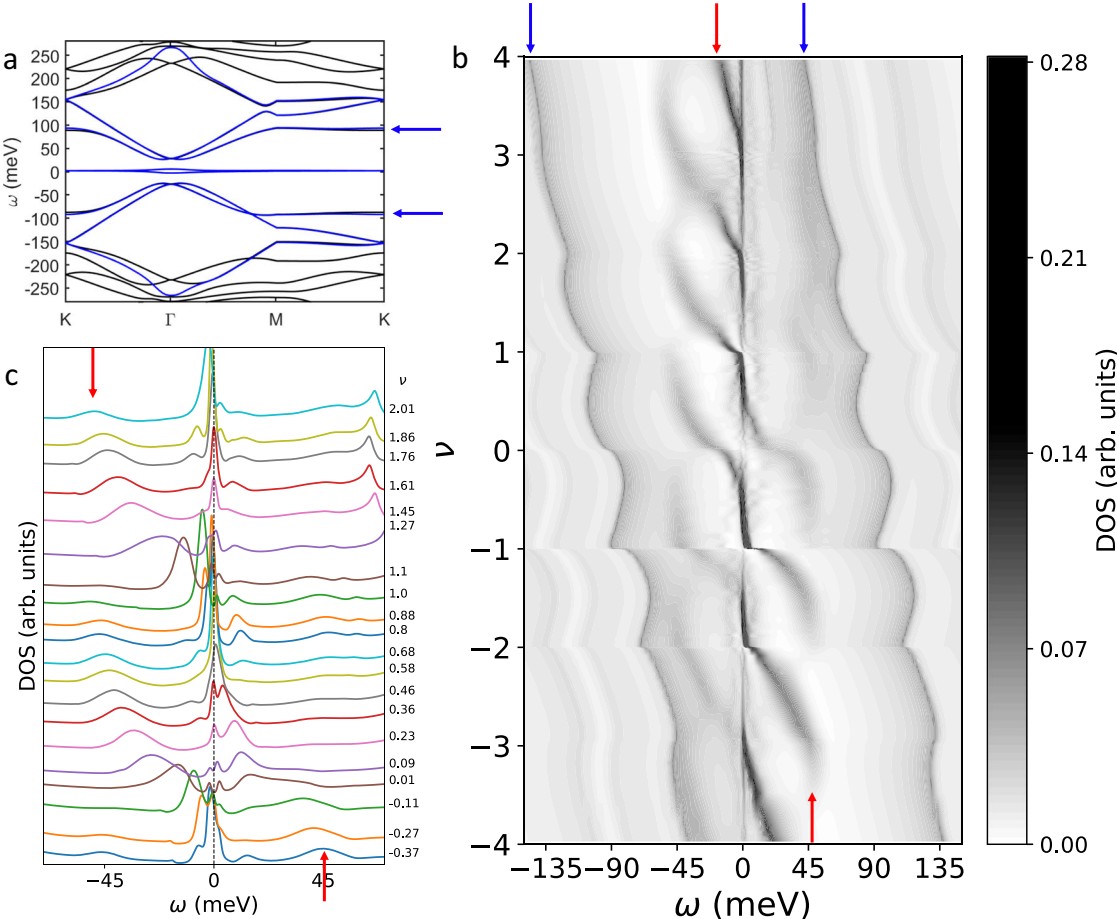

**Fig. 1 | Cascades in the spectral weight. a** Bands of TBG from the continuum model (black) and 8 orbital fitting used in the calculations (blue). **b** Contour plot of the DOS from the DMFT+Hartree calculations versus doping $v$, defined as the number of electrons per moiré unit cell added or removed with respect to the CNP and the energy $\omega$, measured from the Fermi level. Cascades of incoherent spectral weight, resembling STM experiments, flow from $\omega \approx U = 44.5$ meV towards $\omega = 0$ with resets at integer $v$. The cascades at positive and negative energies are shifted in doping.

oscillations. The signatures corresponding to the flat sections of the remote bands are particularly prominent, see blue arrows in (**a**) and (**b**). **c** Line cuts of the DOS shifted vertically for selected $v$ within $-0.5 < v < 2$ with a primarily three peak structure: a quasiparticle peak around $\omega = 0$ and broader Hubbard bands at positive and negative energies, marked with red arrows at the top and lower part of panels (**b**) and (**c**). The Hubbard bands shift with $v$ and merge with the quasiparticle peak at certain dopings. The DOS plotted here includes the whole moiré unit cell. For the DOS with a partial spatial differentiation see Supplementary Fig. 2.

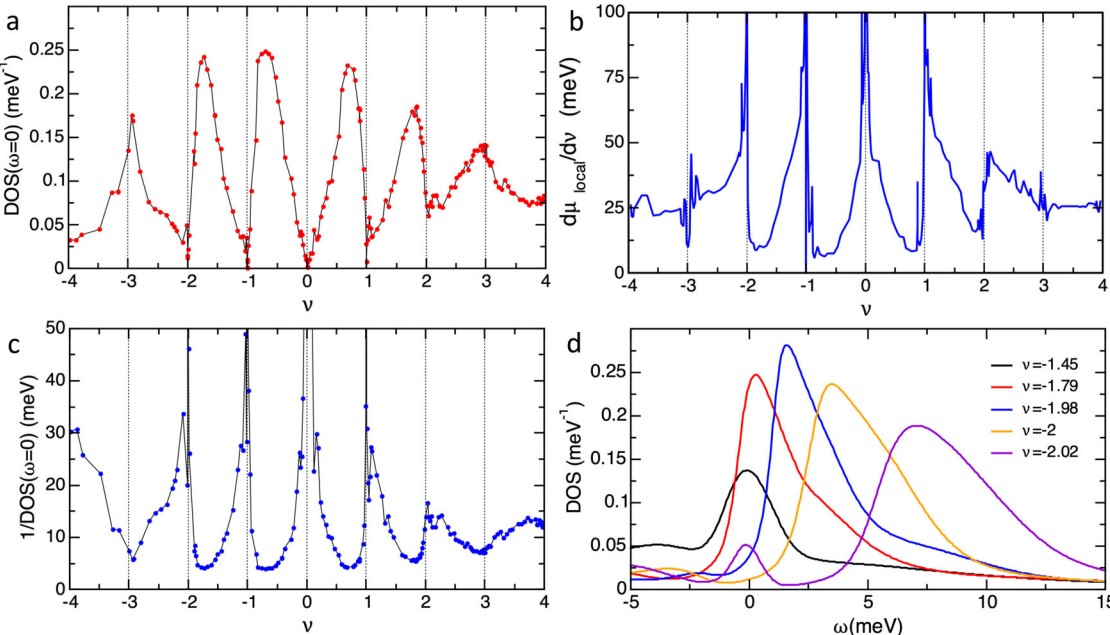

**Fig. 2 | Asymmetric peaks in the inverse compressibility and in the DOS. a** DOS at the Fermi level as a function of doping, showing asymmetric minima at integer fillings within $-2 < v < 2$, except at CNP, where it is symmetric. **b** Local inverse compressibility versus doping with strongly asymmetric peaks. Their shapes strongly resemble the ones in SET measurements[17]. Here $\mu_{local}$ is the chemical potential which enters in the DMFT calculation, see Supplementary Fig. 6. **c** Inverse of the DOS at the Fermi level shown in (**a**) with asymmetric peaks similar to the ones in the local inverse compressibility. **d** Density of states for several dopings close to $v = -2$. All calculations done at $T = 6$ K.

anticross. The cascade flow can also be seen in the line plots within $-0.5 < v < 2$ in Fig. 1c. The three peak structure of the DOS observed in a wide range of dopings, with bumps at positive and negative energies and a quasiparticle peak at zero energy, is common in strongly correlated systems[31]. The broad peaks emerging around $U$, marked with a red arrow in Fig. 1b, c and formed by incoherent spectral weight, constitute the Hubbard bands of the correlated AA$_p$ orbitals, which account for most of the spectral weight of the flat bands. Their unconventional cascade flow is due to their hybridization and interaction with the *lc* orbitals, present at $\Gamma$ of the flat bands (Supplementary Fig. 1), with a finite but very small contribution to the total DOS at low energies, see Supplementary Fig. 2. The strong similarity between the cascades in Fig. 1b, c and the ones in STM experiments[10,20] suggest that the cascades observed experimentally can be explained without invoking symmetry breaking (prohibited in our calculations).

In agreement with STM experiments[10], in Fig. 1b the cascades are accompanied by oscillations of the remote band energies with respect to the chemical potential. These oscillations are best visualized looking at the signatures originating in the flat sections of the remote bands along the M-K direction, marked with blue arrows in Fig. 1a, and in darker grey, around $\pm (50 - 150)$ meV away from the Fermi level, Fig. 1b. These remote band states are dominated by the *lc* orbitals, see Supplementary Figs. 1b and 2b. Adding an electron or hole produces a relative shift between the onsite energies of the AA$_p$ and the *lc* orbitals, which depends on the type of orbital which is doped and on the different interactions. This shift changes the distance between the flat and the remote bands[26]. The remote band peaks oscillate as a function of doping on top of an approximately linear background. The linear contribution is accounted for by the Hartree approximation, Supplementary Figs. 3 and 4, while, as discussed below, the oscillations originate in a non-monotonous filling of the *lc* orbitals with doping. The oscillations are well defined only for $v < |2|$. Beyond this filling the remote bands cross the Fermi level (Supplementary Fig. 5), modifying

the doping dependence of the oscillations and of the DOS($\omega = 0$), which shows neither a minimum at $|v| = 3$ nor a gap at $|v| = 4$. The crossing of the remote bands would happen at larger $v$ in calculations starting from a tight binding model with larger values of $\Delta$ or $\epsilon$. Such crossing does not prevent the AA$_p$ orbitals from being strongly correlated beyond $v > |2|$, as confirmed by the cascade at $n = |3|$ in Fig. 1b and the incoherence signals in Supplementary Fig. 5.

The energy dependence and the strong doping evolution of the DOS in Fig. 1b, with resets at integer fillings, contrast with the much less doping dependent DOS obtained if all the interactions, including $U$, are treated in the Hartree approximation, see Supplementary Fig. 3a. At the Hartree level, the shape of the bands changes[26,39–41] (Supplementary Fig. 3b, c). However, the spectral weight shifted to energies of order $U$ is small, the changes in the density of states are monotonous in doping, and there are no oscillations of the remote bands. The model, including tight-binding parameters and interactions, used in Fig. 1b and in Supplementary Fig. 3a is the same. The only difference between both figures lies in the use of DMFT, instead of Hartree, to treat the intra-moiré unit cell interactions between the AA$_p$ orbitals. The differences in the spectral weight calculated within both approximations show that the formation of the cascades and the oscillations can be explained as an effect of the local correlations, included properly in DMFT.

Figure 2b shows the inverse compressibility as a function of $v$. In spite of some noise, strong asymmetric peaks with maxima at integer fillings are observed for $|v| < 3$ except at CNP, where the peak is approximately symmetric. This sawtooth-like behavior of the inverse compressibility strongly resembles the one observed in SET measurements[17]. The SET experiments were originally interpreted as a signature of ferromagnetic polarization involving Dirac revivals. We emphasize that in our calculations we do not have any ferromagnetic polarization or any other spontaneous symmetry breaking.

An asymmetry with the same sign, peaking around the same values, is found in the inverse of the DOS at the Fermi level, Fig. 2c. Therefore, studying the evolution of the DOS($\omega = 0$) with $v$ can help to

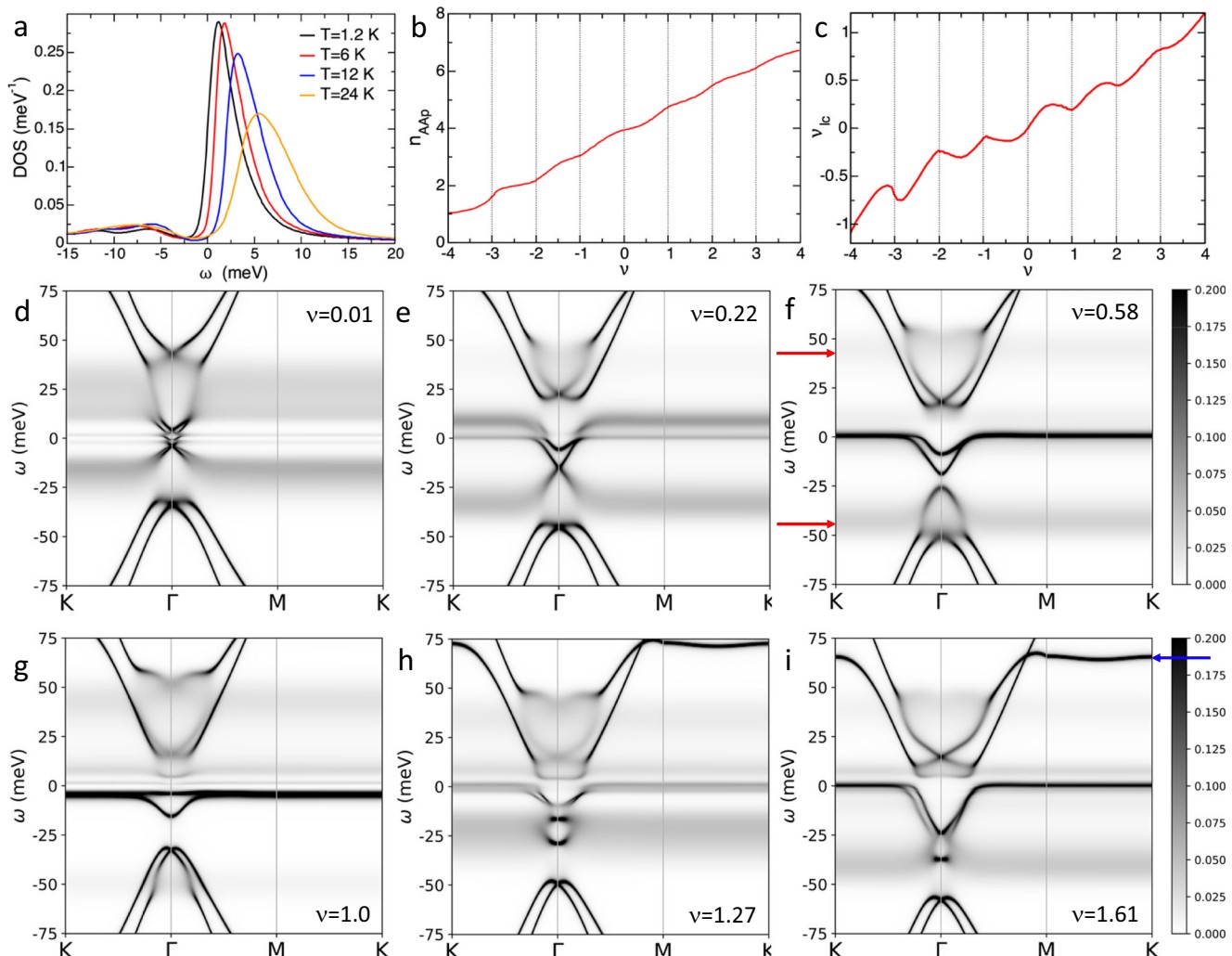

**Fig. 3 | Temperature dependence of the DOS, orbital filling and correlated bands. a** DOS for $\nu = -1.96$ at different temperatures. While the incoherence survives as the temperature rises, the tendency to form a heavy quasiparticle does not. **b** Filling of the $AA_p$ orbitals $n_{AAp}$ with doping, showing a monotonous increase with steps at integer $\nu$. $n_{AAp}$ varies from 0 to 8 accounting for the two types of orbitals, and the valley and spin degeneracy. **c** Doping of the $lc$ orbitals, $\nu_{lc}$, with respect to CNP, including the valley and spin degeneracy. **d−i** Momentum resolved spectral weight respectively for $\nu = 0.01, 0.22, 0.58, 1.0, 1.27$ and 1.61, showing the modifications in the band structure induced by the correlations. Blurred spectra indicate incoherent spectral weight. The resets in the cascades and in the filling at integer $\nu$, not at integer $n_{AAp}$, and the momentum differentiation of the incoherence clarify the role of the hybridization between the $AA_p$ and the $lc$ orbitals and the fragile topology. Red and blue arrows mark, respectively, the Hubbard bands and the flat sections of the remote bands highlighted in Fig. 1. For the specific values of $\epsilon$ and $\Delta$ used here the Hubbard bands overlap with the remote bands. This would not happen for larger gap between the flat and remote bands, $\Delta$, and larger dielectric constant $\epsilon$.

understand the compressibility and its connection with the spectral weight reorganization in the STM measurements. Consistently, the minima in the DOS($\omega = 0$) at integer dopings are strongly asymmetric, except the V-shape at CNP, Fig. 2a.

Figure 2d shows the DOS around $\nu = -2$. Slightly above the integer, for $\nu = -2.02$, it shows a broad peak at $\omega = 7$ meV and a small quasiparticle peak at $\omega = 0$, whose height controls the DOS($\omega = 0$) in Fig. 2a. Close to the integer $\nu$, even tiny changes in doping produce a strong spectral weight reorganization. With decreasing $\nu$, the broad peak becomes narrower, higher, more asymmetric and shifts towards $\omega = 0$. At $\nu = -2$ there is not a quasiparticle peak and the DOS($\omega = 0$) has a minimum. Below this doping the asymmetric peak contributes to the DOS($\omega = 0$), initially through its tail. The maximum DOS($\omega = 0$) is reached around $\nu = -1.7$, slightly above the doping at which the peak is pinned at $\omega = 0$. Beyond this doping, the DOS($\omega = 0$) decreases as the spectral weight is shifted from the quasiparticle peak to a newly formed Hubbard band.

Decreasing temperature from $T = 6$ K to 1.2 K the sharp asymmetric peak at finite $\omega$ slightly shifts towards $\omega = 0$, Fig. 3a and Supplementary Fig. 7. Correspondingly, the maximum in the DOS($\omega = 0$) and the minimum in the inverse compressibility will appear slightly closer to the integer. We find pseudogaps in the DOS($\omega = 0$) at $\nu = 1, -1$ and −2, which get smaller as the temperature is reduced, suggesting that they could close at $T = 0$. However, we emphasize that the physics discussed here is not linked to whether there is a pseudogap or a small gap at exactly integer fillings but to a reorganization of the spectral weight at all dopings of the flat bands in a large energy range, with incoherent bands starting around $U$ and progressively approaching $\omega = 0$. When $T$ increases, the peak shifts to higher energies, becomes broader, more symmetric and smaller in height. Close to $\nu = -2$, the height is reduced approximately to half of its value when $T$ rises from 6 K to 24 K. Notably, the spectrum does not return to its non-interacting or Hartree shape at high temperatures, Supplementary Fig. 7. The incoherent spectral weight is resilient with temperature, becoming

broader and less defined around integer values. This is consistent with experimental findings of broadened cascade signatures up to at least 50 K[17,18,20].

The cascade dependence on $\omega$ and $\nu$ originates in the tendency of the $AA_p$ orbitals to form local moments and become incoherent, and their hybridization and interactions with the $lc$ orbitals which accomodate part of the doping and tend to screen the local moments. As shown in Supplementary Fig. 8, in the absence of the $lc$ electrons, at integer $\nu$ the $AA_p$ orbitals would be Mott insulators with large gaps, their spectral weight becoming incoherent and shifted to the Hubbard bands. Doping such Mott insulators would drive them metallic with a heavy quasiparticle at the Fermi level. Hole (electron) doping creates the quasiparticle in the lower (upper) Hubbard band[32]. Therefore, even in the absence of the $lc$ orbitals, resets in the shape and position of the Hubbard bands would appear around integer $\nu$ but the reorganization of the spectral weight in this case is different to the cascades in Fig. 2b which arise when the hybridization and interactions between the $AA_p$ and the $lc$ orbitals are included.

Figure 3b shows how the $AA_p$ orbitals accommodate most of the doping, with weak steps at integer $\nu$, while the $lc$ orbitals (Fig. 3c) show a non-monotonic occupation as a function of the total doping. This behaviour contrasts with both the filling of the $AA_p$ orbitals in the absence of the $lc$ orbitals (all the doping would go to the $AA_p$ orbitals with no steps) and with simpler heavy fermion systems, with hybridization between the $AA_p$ and the $lc$ orbitals but including only the on-site interaction U between the $AA_p$ orbitals. In the latter case, pronounced steps are expected in both the $AA_p$ and the $lc$ orbitals and the non-monotonic dependence of the $lc$ filling would be absent. The behaviour shown in Fig. 3b–c can be understood as follows: when adding carriers from the CNP, first the $lc$ orbitals are doped up to a value at which the $AA_p$ orbitals starts accepting carriers. Due to the interactions between the $AA_p$ and the $lc$ orbitals, once doping the $AA_p$ orbitals becomes energetically possible, the $lc$ orbitals are partially emptied to reduce the cost of adding carriers to the $AA_p$ orbitals. At a larger doping, there is a plateaux in the $AA_p$ filling and it becomes energetically favourable to dope the $lc$ orbitals again. The oscillations of the remote bands with respect to the chemical potential follow this non-monotonic occupation of the $lc$ orbitals, Fig. 3c. The resets happen at integer doping as a consequence of the hybridization between the $AA_p$ and the $lc$ orbitals.

Due to the fragile topology of TBG, the $lc$ electrons contribute to the flat bands around $\Gamma$, Supplementary Fig. 1. The different interactions in both types of orbitals result in band bending around $\Gamma$ when the system is doped[26,39–41], see Fig. 3d–i, and Supplementary Fig. 3. Our DMFT + Hartree calculations predict effects that can be measured in future experiments with nano-ARPES[42,43] or the Quantum Twisting Microscope[44]. Following the step-wise and non-monotonic orbital fillings, the bending of the bands around $\Gamma$ is not proportional to doping, but produces band flattening around certain fillings, see Fig. 3g. More remarkably, we predict a strong momentum selective incoherence. Due to the strong correlations, the $AA_p$ orbitals form local moments. Associated to the moment formation, the spectral weight of these orbitals is suppressed at small energies and shifted to the broad and incoherent Hubbard bands, marked with a red arrow in Fig. 3f. Because the $AA_p$ orbitals are absent at $\Gamma$ in the flat bands, the incoherence is strongly reduced around this momentum. The momentum selective incoherence is particularly visible in Fig. 3d, e. At fillings below integer dopings, the incoherence at low energies is suppressed due to the formation of heavy quasiparticles except at the CNP where the screening of the local moments is not being operative.

The temperature dependence of the DOS in Fig. 3a is consistent with this picture. In the calculations, the screening of the local moment is not complete. As the temperature is reduced, the high peak at integer values approaches the Fermi level towards the formation of a heavy quasiparticle. With increasing temperature, the screening of the local moment is reduced as the entropy of the local moment is higher than the Fermi liquid one. The formation of the heavy quasiparticles happens only at low temperatures while the local moment physics, responsible for the incoherence, remains at high temperatures[18,19].

## Discussion

In conclusion, we have shown that the experimentally observed cascade phenomenon can be explained without invoking symmetry breaking orders as a consequence of the reorganization of spectral weight induced by the strong correlations. Similar to what happens in Mott-like systems, the spectral weight becomes incoherent as the correlated $AA_p$ electrons form local moments. The hybridization between the $AA_p$ orbitals to the less correlated electrons, due to the fragile topology of TBG, results in momentum selective incoherence and promotes the formation of a heavy quasiparticle around integer dopings. The cascades connect TBG with other strongly correlated systems such as high temperature superconductors, heavy fermion systems and oxides through their hallmark: the incoherent spectral weight. In our description, the cascades constitute the *normal* state in which low temperature symmetry breaking transitions such as ferromagnetism, Chern insulators, superconductivity, intervalley coherence or nematicity can emerge at low temperatures. The cascade phenomenon originates in the local correlations induced by the intra-moiré unit cell interaction $U$ between the $AA_p$ orbitals. This interaction is not much screened by gates farther than 5 nm[26]. Therefore, the cascades are expected to survive to the proximity of gates above this distance, contrary to other effects controlled by longer range interactions. This dependence can help identifying the different phenomena.

Cascades similar to the ones discussed here have been recently observed in twisted trilayer graphene[45] for which a heavy fermion model has been also proposed[46,47] and we expect them to be described by the same mechanism. On the other hand, this description is not applicable to the signatures detected in the compressibility, also named cascades, in graphene ABC trilayer and Bernal bilayer without moiré patterns[48–50]. A symmetry breaking origin of these signatures is consistent with differences in the shape, doping dependence, and the much lower temperature range in which they appear. Notwithstanding, a strong reorganization of the spectral weight, related to the one discussed here, with heavy quasiparticles but without the effects of the fragile topology, is expected if the ABC trilayer is aligned with hBN forming a moiré lattice[51].

When we were completing this manuscript some related works appeared on the arXiv[52–54].

## Methods

We start from the bands of a $\theta = 1.08°$ TBG calculated using the continuum model, with interlayer tunnelling ratio between the AA and AB stacking regions $w_0/w_1 = 0.78$ and keeping the $\sin(\theta/2)$ term, which slightly breaks particle-hole symmetry[35–37]. The ratio $w_0/w_1$, which accounts partially for relaxation effects, determines $\Delta$, the gap between the flat and the remote bands at $\Gamma$. For a given angle, $\Delta$ is larger for smaller $w_0/w_1$. The value $w_0/w_1 = 0.78$ has been chosen on the basis of previous theoretical proposals[37]. We adapt the 8 orbital model per valley and spin from[29,38] to fit the low energy bandstructure. The overlap between the eigenstates at the flat bands in the Wannier model used here and in the fitted continuum model is above 99%. The Wannier functions have to be recalculated for each angle. The Wannier states have length scales comparable to the moiré unit cell, not the graphene atomic lattice, and interaction scales of the order of tens of meVs, see Supplementary Fig. 1d. We include density-density interactions up to 7 moiré lattice

constants and neglect the considerably smaller Hund's coupling, pair-hopping and exchange interactions[26]. The value of the dielectric constant $\epsilon = 12$ was estimated in ref. [26] by comparing the deformation of the bands produced by the Hartree approximation in the Wannier model used here and in atomic calculations which include the electronic states up to very high energies[41]. The choice of the temperature $T = 6$ K was inspired by experimental results. Experimentally, at this temperature the cascade signatures are well pronounced and the features associated with the low temperature symmetry breaking states are mostly absent. In our calculations the symmetry-breaking transitions have been suppressed. We do not claim that at this exact temperature all the symmetry breaking states have disappeared. The critical temperature of the low temperature states could depend to some extent on the twist angle and the filling. Based on the interactions and bandwidth magnitudes[26] we distinguish between strongly correlated $AA_p$ ($p_+$ and $p_-$) and less correlated $lc$ orbitals: (AA $s$, AB $p_z$ and DW $s$). DMFT[32,33] is used on the onsite interaction $U$ among the $AA_p$ orbitals, which acquire a frequency dependent self energy $\Sigma(\omega)$. Single site DMFT deals with local correlations exactly via $\Sigma(\omega)$, while neglecting the momentum dependence of the self-energy, i.e. the non-local correlations. The non-local correlations become less important with increasing dimensionality. In TBG, with an $AA_p$ triangular lattice, the effective dimensionality given by the coordination number is larger than in square lattice materials, such as iron superconductors, heavy fermion systems and oxides where DMFT has been successfully applied. The rest of the interactions, including intersite ones, are treated at the Hartree level. In this approximation, their effect enters through onsite potentials which modify the band shape producing for instance the bending of the flat bands. The $lc$ orbitals also get a frequency dependent self-energy through the hybridization with the $AA_p$ orbitals. In the DMFT calculation, the $lc$ orbitals enter via an effective hybridization dependent on the Hartree onsite potentials. A double step self-consistency loop is implemented. A number of DMFT iterations are run starting from a given set of Hartree onsite potentials and avoiding double counting the interaction U. As an outcome of the DMFT, the self energy, the Green's function and the charge of the $AA_p$ orbitals are calculated. Due to the hybridization between the two types of orbitals, the orbital filling of all the orbitals is modified. The Hartree onsite shifts are re-calculated with the new fillings and new DMFT iterations are run with the re-calculated onsite potentials. Convergence is reached when the Green's functions, self-energy and orbital fillings do not change between either DMFT or Hartree iterations. To avoid double counting the interaction effectively included in the tight binding model parameters, we sustract the onsite potentials obtained at the CNP in the Hartree approximation[40]. The single-site DMFT calculations are performed at a given chemical potential and finite temperature using a continuous time quantum Monte Carlo impurity solver[55] as implemented in ref. [56]. The density is obtained to an accuracy ≈0.01 electrons. Symmetry breaking is prohibited imposing equal onsite potentials, self-energies and Green's functions for equivalent orbitals. We use the local chemical potential $\mu_{local}$ which enters into the DMFT part of the self-consistency loop to calculate the inverse compressibility, see Supplementary Fig. 6. To calculate the spectral weight we perform the analytic continuation of the Matsubara self-energy using the maximum entropy method[57].

## Data availability
The data plotted in the figures included in the main text or in the Supplementary Information have been deposited in Zenodo Repository and are available in Open Access in https://doi.org/10.5281/zenodo.8079291.

## Code availability
Codes required for reproducibility are available upon request to the authors.

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

## Acknowledgements

We thank M. Rozenberg, M. Civelli, E. Miranda, M.C.O. Aguiar, A. Yazdani, A.V. Chubukov, F. de Juan and B.A. Bernevig. M.J.C, E.B. and A.D. acknowledge funding from PGC2018-097018-B-I00 (MCIN/AEI/FEDER, EU), M.J.C. and E.B. acknowledge PID2021-125343NB-100 (MCIN/AEI/FEDER, EU). A.C. acknowledges support from UBACyT (Grant No. 20020170100284BA) and Agencia Nacional de Promoción de la Investigación, el Desarrollo Tecnológico y la Innovación (Grant No. PICT-2018-04536). A.D. acknowledges the French National Research Agency (TWIST- GRAPH, ANR-21-CE47-0018).

## Author contributions

E.B. and M.J.C. conceived the project. A.D. adapted and implemented codes to compute the Wannier functions, the interactions between them and calculations in the Hartree approximation. E.B. and A.C. designed the combined DMFT+Hartree calculation and A.C. implemented it. A.D. and M.J.C. run the DMFT+Hartree calculations with punctual assistance from other authors. All the authors participated in the analysis and the discussion of the results. M.J.C., A.C., and A.D. prepared the figures. E.B. supervised all the project and wrote the manuscript with contributions from all the authors.

## Competing interests

The authors declare no competing interest.
