## [Peer Review File · Nature Communications]

REVIEWER COMMENTS

Reviewer #1 (Remarks to the Author):

In this work, the authors provided a convincing alternative explanation of experimentally observed cascades in twisted bilayer graphene, which is a very intense area of research right now, of broad and general interest. It is appropriate for the readers of the Nat. Comm. journal and I recommend the paper be published upon answering the following questions.

1. The authors claim, which strong theoretical support, that cascades can be observed due to strong electronic correlations alone. However, since they, as they profess, never include symmetry-breaking effects like ferromagnetism, and then to claim they "cascade phenomenon is not associated to a symmetry-breaking" is presumptive and affects the overall integrity of the paper. The latter claim can e.g., be made by including symmetry-breaking effects but no strong electronic correlations, and then observing no cascades. Supporting experiments differentiating that the two mechanisms (strong correlations vs symmetry breaking) can/should then be supplied to further bolster what is a strong claim.

In other words, as it stands, the best claim the authors can make is that strong electronic correlation is also a plausible alternative explanation for cascades, that is previously not considered, but symmetry-breaking can still not be ruled out.

2. The authors added nonlocal hartree interaction in addition to the local DMFT interactions to account for nonlocal and long-range effects. These long-range effects (e.g., coulomb interactions) are almost always screened by electrons in 2D materials (e.g., via RPA and nonlocal vertex corrections e.g., GW or T-matrix diagrams). Screening effects renormalizes the band energy by orders of $\sim 1-10$ eV. Since the authors are interested in electronic effects that is smaller and in order of magnetudes (\sim meV), will these screening/vertex effects in some phase space (e.g., twist angles or fillings) be important and possibly dominates the strong-correlation features as studied by the authors in this paper?

3. The authors fit the low-energy bandstructure of $\theta=1.08$ deg. Nonetheless, while fitting bandstructure guarantees that the eigen-energies of the fitted states matches the eigen-energies of the actual states, there is no guarantee that the eigen-functions match. Furthermore, there is no further guarantees that eigen-functions should still be correct at all angles. Moreover, what about the high-energy states that should and will contribute to the nonlocal selfenergy as well? Additionally, the fitted $U=44.5$ meV is small, compared to e.g., coulomb interactions. What is the error bar of the fitting, how robust/sensitive is this result to the value of U and are there phase spaces are other energy scales where other interactions dominate?

4. "Hole (electron) doping creates the quasiparticle in the lower (upper) Hubbard band[27]. Therefore resets in the shape and position of the Hubbard bands appear around integer." Why can't fractional doping creates satellites features that has similar effects since strong correlations are important and central (analogous to fractional quantum hall effects which is integer quantum hall effects in the presence of strong correlations)

Reviewer #2 (Remarks to the Author):

In the submitted manuscript entitled, "Heavy quasiparticles and cascades without symmetry breaking in twisted bilayer graphene", the authors study the effects of interactions on the many-body ground state of the twisted bilayer graphene (TBG) using dynamical mean field theory (DMFT) together with Hartree corrections. The authors, in particular, focus on the "cascade" present in TBG and explain its origins within the scope of their analysis. The claimed novelty of the work lies in explaining the origins of the "cascade" without reference to symmetry breaking.

While I find the manuscript exciting and of high relevance to ongoing research in the moiré field, I have reservations regarding the authors' proposed explanation of the cascade and how it differs from the existing literature (I expand on my reasoning below). Until the authors clearly and convincingly answer my questions and modify the manuscript accordingly, I cannot make a recommendation for this paper to be published in a high-profile journal such as Nature Communications in its present form. If the authors can provide convincing arguments, then I am happy to recommend its publication.

Before diving into the discussion of the authors' key claim, I have the following suggestions regarding the manuscript, which, I believe, will help with the readability of the manuscript and/or authors' claims. These are:

- Could the authors provide at least a one-paragraph in-text introduction to the dynamical mean field theory and explain in what way this method is superior to Hartree-Fock, DMRG, strong-coupling calculation, or other approaches? What is its shortcoming as well? Here this point is specifically important since the authors intend to change the accepted and experimentally verified paradigm of the cascade description in the moiré systems (which arises in all of these other calculations), the burden falls on the authors to demonstrate why and how these methods, in the authors' opinion, fail and the DMFT point of view does not.

- I invite the authors to improve the form of the in-text citations to indicate at which point the idea was already present and discussed in the literature and at which point the view is the authors' invention. One example is the discussion in lines 18-20. Is it the authors' own idea that the physics of cascade constitutes the parent state from which the correlated phenomena emerge? If so, then I'm afraid I have to disagree, and it was introduced in Ref. 17, for example. As such, I would expect the citation to be present either after "suggesting" or at the end of the sentence in line 20. There are several more examples like that throughout the text, which the reader can misinterpret.

- The paragraph starting on line 39, in my opinion, should be expanded. Specifically, the authors should discuss their motivation or physical origin of different parameters $\epsilon = 12$ K, $T = 6$ K, the form of the Coulomb interaction, E.g., I imagine authors choose $T = 6$ K as it is the temperature scale at which cascade is most pronounced, but other correlated states do not yet start to manifest.

- In Fig. 1b, to help readers, I would invite the authors to annotate features on the DOS diagram.

- The language throughout the manuscript needs to be sharpened and improved. Specifically, I caught a few typos, such as "cascade phenonema" in the abstract, or imprecise statements such as "first slowly and a bit faster" or "a large amount of spectral weight is visible".

- A highly confusing part is the comparison of the Fig. 3 momentum resolved spectral weight against the Fig. 1c. For example, looking for instance at $\nu = 0.58$, assuming that $\omega = 0$ corresponds to chemical potential, the peak of the DOS at $\omega = -45$ or $\omega = 45$ meV would originate from the incoherent spectral weight. These features occur in momentum states that I would otherwise attribute to dispersive bands, c.f. Fig. 1a. Is that correct? If not, where in the states would come from dispersive bands? If indeed these are the dispersive bands, then this is in sharp disagreement with STM works (see works from Yazdani or Nadj-Perge groups) that can track the van Hove singularities of the two flat bands and their position wrt to the dispersive bands. The authors should clarify the features of these momentum-resolved spectral plots and discuss how they connect to existing experimental works.

Let me now proceed to my main worry - the authors claim that the symmetry-breaking process does not explain cascade physics, and instead, it is explained by local correlations that are ONLY correctly described by the DMFT. I recommend that authors remove or completely modify the discussion in lines 90-94, 58-59, and the abstract. Symmetry-breaking transition as described by many works on Hartree-Fock, Strong-Coupling, or DMRG calculations from Zaletel, Vishvanth, Vafeek, Bernevig, and many others (none of which the authors discuss or cite), explain or are at least consistent with the experimental observations. The authors, thus, cannot discard them and

say that their DMFT description is superior. In fact, the language of symmetry breaking naturally explains recent experiments (see the Yazdani paper on the observation of IKS and TIVC states, for example), which the authors' treatment does not naturally lend itself to.

Moreover, the authors do not provide any experimental prediction of their results, just postdiction, which are not different in any experimentally measurable way from those of Hartree-Fock, Strong-Coupling, or DMRG works. As such, it becomes a matter of belief for the authors what is "correct" unless a precise, experimentally measurable difference can be identified. I would therefore recommend that authors rephrase the discussion and the whole message of the manuscript as not invalidating the symmetry-breaking transition point of view, but rather providing an alternative, but equally valid, description.

To summarize, I enjoyed the authors' manuscript immensely. I would be more than happy to recommend its publication if the authors can make their claim about symmetry breaking more explicit. This authors can do this by either invaliding or providing clear and precise experimental predictions of how authors' description of the cascade can be differentiated from that given through the symmetry-breaking transition, which naturally emerges through the Hartree-Fock, Strong-Coupling, and the DMRG picture.

Reviewer #3 (Remarks to the Author):

In the manuscript under review, the authors investigate the cascade phenomenon of the parent state in twisted bilayer graphene (TBG) in terms of self-consistent DMFT+Hartree calculations. They find that the cascade phenomenon observed in spectroscopic and compressibility measurements can be explained well without spontaneous symmetry breaking, but via local moment formation and heavy-fermion physics. This is a very interesting and timely result, and the authors provide new insights that they are able to obtain due to their advanced modelling and methodology. The results are also of interest beyond TBG for strongly correlated electron systems in general. Thus, I recommend publication, in principle, if the following issues are addressed upon resubmission.

1) The authors show that the cascade phenomenology in TBG is consistent with a strong modification of the normal state without symmetry breaking. At the same time, previous publications cited in the manuscript show that alternatively the cascade is consistent with models involving symmetry breaking transitions. Symmetry breaking is suppressed by hand in the used method. What would happen if symmetry breaking was allowed? Can the authors make clearer if their explanation is more than "just" an alternative or how to test the different scenarios experimentally.

2) A cascade of transitions has been observed in less correlated graphene systems like Bernal bilayer and rhombohedral trilayer graphene. One interpretation is that analogous physics, but on different interaction scales, is responsible for the similarity of the observations in different graphene systems. Why do the authors think that this is not the case and TBG has to be considered differently?

3) The authors use the notion of hybridization/coupling between the narrow and remote bands in TBG interchangeably with the notion of fragile topology, and thus argue that fragile topology is important for the cascade phenomenology. However, hybridization between these bands and fragile topology are not the same thing. There can be hybridization in models without fragile topology. Since the authors do not show a comparison to a model with hybridization but without fragile topology, how can they tell which effect is the decisive one?

4) In a similar context, it would be very helpful if the authors could explain the relation to the heavy-fermion model of Ref. 23 which was used in Refs. 36-38 to study the cascade phenomenology in TBG. Do the results obtained from different models and methods agree? Why (not)? What would be a minimal model? How important are the long-ranged interactions used here?

5) As a minor point: figures in Fig. 1 and Fig. 3 show many features each, so that it can be difficult to follow the discussion of these features in the text. It could be helpful to add some descriptors/arrows into the figures themselves.

RESPONSE TO REVIEWERS' COMMENTS

In the following, we answer the reviewer comments point by point (referees' comments in blue and our answers in black) and detail the main changes in the manuscript (marked in red in the pdf). We have also added 16 new references:

- 18 Y. Saito, F. Yang, J. Ge, X. Liu, T. Taniguchi, J. Watanabe, Kenji ang Li, E. Berg, and A. F. Young, Isospin pomeranchuk effect in twisted bilayer graphene, *Nature* 592, 7853 (2021).
- 19 A. Rozen, J. M. Park, U. Zondiner, Y. Cao, D. Rodan-Legrain, T. Taniguchi, K. Watanabe, Y. Oreg, A. Stern, E. Berg, P. Jarillo-Herrero, and S. Ilani, Entropic evidence for a pomeranchuk effect in magic-angle graphene, *Nature* 592, 214 (2021).
- 21 J. Kang, B. A. Bernevig, and O. Vafek, Cascades between light and heavy fermions in the normal state of magic-angle twisted bilayer graphene, *Phys. Rev. Lett.* 127, 266402 (2021).
- 23 J. P. Hong, T. Soejima, and M. P. Zaletel, Detecting symmetry breaking in magic angle graphene using scanning tunneling microscopy, *Phys. Rev. Lett.* 129, 147001 (2022).
- 30 H. Shi and X. Dai, Heavy-fermion representation for twisted bilayer graphene systems, *Phys. Rev. B* 106, 245129 (2022).
- 34 E. Pavarini, E. Koch, A. Lichtenstein, and D. Vollhardt, Dynamical Mean-Field Theory of Correlated Electrons, *Lecture Notes of the Autumn School on Correlated Electrons 2022* (Verlag des Forschungszentrum Julich, 2011).
- 41 Z. Goodwin, V. Vitale, X. Liang, A. Mostofi, and J. Lischner, Hartree theory calculations of quasiparticle properties in twisted bilayer graphene, *Electronic Structure* 2, 034001 (2020).
- 42 M. Cattelan and N. Fox, A perspective on the application of spatially resolved arpes for 2d materials, *Nanomaterials* 8, 284 (2018).
- 43 S. Lisi, X. Lu, T. Benschop, T. A. de Jong, P. Stepanov, J. R. Duran, F. Margot, I. Cucchi, E. Cappelli, A. Hunter, A. Tamai, V. Kandyba, A. Giampietri, A. Barinov, J. Jobst, V. Stalman, M. Leeuwenhoek, K. Watanabe, T. Taniguchi, L. Rademaker, S. J. van der Molen, M. P. Allan, D. K. Efetov, and F. Baumberger, Observation of flat bands in twisted bilayer graphene, *Nature Physics* 17, 189 (2021).
- 44 A. Inbar, J. Birkbeck, J. Xiao, T. Taniguchi, K. Watanabe, B. Yan, Y. Oreg, A. Stern, A. Berg, and S. Ilani, The quantum twisting microscope, *Nature* 614, 682 (2023).
- 45 H. Kim, Y. Choi, C. Lewandowski, A. Thomson, Y. Zhang, R. Polski, K. Watanabe, T. Taniguchi, J. Alicea, and S. Nadj-Perge, Evidence for unconventional superconductivity in twisted trilayer graphene, *Nature* 606, 494 (2022).
- 46 A. Ramires and J. L. Lado, Emulating heavy fermions in twisted trilayer graphene, *Phys. Rev. Lett.* 127, 026401 (2021).
- 47 J. Yu, M. Xie, B. Bernevig, and S. Das Sarma, Magic-angle twisted symmetric trilayer graphene as topological heavy fermion problem, [arXiv:2301.04171 \[cond-mat.str-el\]](https://arxiv.org/abs/2301.04171) (2023).
- 48 H. Zhou, T. Xie, A. Ghazaryan, T. Holder, J. R. Ehrets, E. M. Spanton, T. Taniguchi, K. Watanabe, E. Berg, M. Serbyn, and A. F. Young, Half and quarter metals in rhombohedral trilayer graphene, *Nature* 598, 429 (2021).
- 49 H. Zhou, L. Holleis, Y. Saito, L. Cohen, W. Huynh, C. L. Patterson, F. Yang, T. Taniguchi, K. Watanabe, and A. F. Young, Isospin magnetism and spin-polarized superconductivity in bernal bilayer graphene, *Science* 375, 774 (2022), <https://www.science.org/doi/pdf/10.1126/science.abm8386>.
- 50 S. C. de la Barrera, S. Aronson, Z. Zheng, K. Watanabe, T. Taniguchi, Q. Ma, P. Jarillo-Herrero, and R. Ashoori, Cascade of isospin phase transitions in bernal-stacked bilayer graphene at zero magnetic field, *Nature Physics* 18, 771 (2022).

REVIEWER COMMENTS AND RESPONSES

Reviewer #1 (Remarks to the Author):

In this work, the authors provided a convincing alternative explanation of experimentally observed cascades in twisted bilayer graphene, which is a very intense area of research right now, of broad and general interest. It is appropriate for the readers of the Nat. Comm. journal and I recommend the paper be published upon answering the following questions.

We thank the referee for considering that our explanation of the cascades is convincing and the work is appropriate for the readers of Nature Communications.

1. The authors claim, which strong theoretical support, that cascades can be observed due to strong electronic correlations alone. However, since they, as they profess, never include symmetry-breaking effects like ferromagnetism, and then to claim they "cascade phenomenon is not associated to a symmetry-breaking" is presumptive and affects the overall integrity of the paper. The latter claim can e.g., be made by including symmetry-breaking effects but no strong electronic correlations, and then observing no cascades.

Strong correlation effects are a consequence of the ratio between the local interaction U and kinetic energies W . The large energies involved in the cascade phenomena (several tens of meV, larger than the bandwidth) and the high temperatures at which the cascades are observed indicate that the ratio U/W is large and hence that correlations cannot be neglected. We have checked that the cascades remain well formed even for considerably smaller values of U/W than the one in the manuscript, see below.

Our proposal is that the cascades are already present in the normal state and, therefore, there is no need to invoke extra reasons, such as symmetry breaking, to explain them. This doesn't mean that symmetry breaking effects are not important to explain certain low temperature effects (e.g. insulating gaps, anomalous Hall effect or charge modulation) to be differentiated from the cascades.

Supporting experiments differentiating that the two mechanisms (strong correlations vs symmetry breaking) can/should then be supplied to further bolster what is a strong claim.

Symmetry breaking states have been detected experimentally at low temperatures. As mentioned before, the low temperature orders are to be differentiated from the cascades which survive up to very high temperatures and involve larger energy scales, see further discussion in the response to the second referee.

An indirect evidence of the importance of strong correlations is the small energy difference between the variety of symmetry breaking orders that have been theoretically studied (~ 0.1 meV in many cases), questioning the presence of a particular symmetry breaking order at 50 K. This suggests that the main driving force of the spectral weight reorganization and its survival at high temperatures is not a symmetry breaking order but the formation of local moments that we propose. These local moments would constitute the parent state from which the low temperature ordered state arises.

In our work, besides describing existing experiments, we predict momentum selective incoherence (larger incoherence away from Γ) that can be detected in future experiments like nano-ARPES, or the newly developed Quantum Twisting Microscope. In the new version

we emphasize this and other predicted effects such as a non-monotonicity in band bending with doping (lines 145-156). In future work we will study the effect of strong correlations in other type of measurements which can distinguish our proposal from symmetry breaking ones.

In other words, as it stands, the best claim the authors can make is that strong electronic correlation is also a plausible alternative explanation for cascades, that is previously not considered, but symmetry-breaking can still not be ruled out.

For clarity in our message we have made changes in several paragraphs of the manuscript indicating that symmetry breaking is not required to explain the cascades because these features are expected to be present already in the normal state as a consequence of strong correlations. Related text has been changed in the abstract, lines 65-67 (page 3), and in the conclusions.

We have also emphasized in page 6 that the momentum selective incoherence is a new prediction which can be experimentally detected.

2. The authors added nonlocal hartree interaction in addition to the local DMFT interactions to account for nonlocal and long-range effects. These long-range effects (e.g., coulomb interactions) are almost always screened by electrons in 2D materials (e.g., via RPA and nonlocal vertex corrections e.g., GW or T-matrix diagrams). Screening effects renormalizes the band energy by orders of $\sim 1-10$ eV. Since the authors are interested in electronic effects that is smaller and in order of magnetudes (\sim meV), will these screening/vertex effects in some phase space (e.g., twist angles or fillings) be important and possibly dominates the strong-correlation features as studied by the authors in this paper?

The interactions are calculated following a previous paper (PRB 102, 155149 (2020)), by part of the authors of this work, where all the possible interactions were addressed, as well as the effect of nearby gates. The results and energy scales obtained in that work compared well with a previous analysis of the screening of interactions in the continuum model for TBG using constrained RPA (Phys. Rev. B 100, 161102 (2019)). From comparison with DMFT+GW calculations in simpler systems (Aryal et al, PRB 87, 125149 (2013)) we expect that the remaining screening effects which may emerge within the effective Wannier model would be small, only a slight renormalization of the interaction, and would not play a significant role.

Please note that the relevant interaction scales in TBG are of the order of tens of meVs as we are not dealing with a model for the graphene carbon atoms (in which the hopping terms and the Coulomb energies are in the range of eV) but effective Wannier functions in the moiré lattice, with thousands of atoms in the unit cell. In such a model, the hopping and interactions are in the meV range.

For clarification we have introduced some sentences about the energy scales and screening in the methods section: lines 198-199, 201-203.

3. The authors fit the low-energy bandstructure of $\theta=1.08$ deg. Nonetheless, while

fitting bandstructure guarantees that the eigen-energies of the fitted states matches the eigen-energies of the actual states, there is no guarantee that the eigen-functions match. Furthermore, there is no further guarantees that eigen-functions should still be correct at all angles.

We have chosen the Wannier functions in such a way that they fulfill the symmetry requirements of TBG. These Wannier functions have to be recalculated for each angle. We have checked that the overlap between the eigenstates at the flat bands in our effective Wannier model and in the fitted continuum model is above 99%.

We have introduced clarifying sentences in the methods section: lines 196-197.

Moreover, what about the high-energy states that should and will contribute to the nonlocal selfenergy as well?

In the previous work (PRB 102, 155149 (2020)) some of us estimated the value $\epsilon=12$ for an isolated twisted bilayer graphene by comparing the band deformation obtained in our Wannier model in the Hartree approximation with the deformation obtained in Goodwin et al, Electron. Struct. 2, 034001 (2020) using the same approximation and an atomistic model which included a large number of bands. In this way, the effect of the high-energy states which are not kept in the effective model but could contribute to the screening of the interactions is introduced in the effective dielectric constant ϵ .

Additionally, the fitted $U=44.5$ meV is small, compared to e.g., coulomb interactions. What is the error bar of the fitting, how robust/sensitive is this result to the value of U

As mentioned above, the value of U should not be compared with the interactions between electrons in each carbon atom as we are dealing with orbitals with a length scale related the moiré unit cell.

We note that in this work we have not done any fitting of the interactions to compare with the experiment. The value of $U=44.5$ meV has been calculated with the effective Wannier orbitals and assuming a $1/r$ interaction and the effective dielectric constant $\epsilon=12$ obtained previously.

We have introduced a sentence in the methods section: lines 201-203.

We note that the emergence of the cascades is not a consequence of the specific interactions that we have selected. Here we send some new results which show that the cascades remain well formed in the normal state at interactions ($U=26.7$ meV corresponding to $\epsilon=20$) considerably smaller than the ones used in the manuscript ($U=44.5$ meV corresponding to $\epsilon=12$).

Figure 1. Line plots of the density of states (DOS) at $U=26.7$ meV in the range of fillings $-0.5 < \nu < 2.5$. Left and right plots show the results in different energy range to highlight the resets close to the Fermi level (left) and the oscillations of the remote band peaks (right).

and are there phase spaces are other energy scales where other interactions dominate?

We do not expect that other interactions alter the picture for the cascades described here. Some quantitative changes of the quasiparticle band shape could emerge if the long-range part of the interaction is screened by nearby gates or if the other interactions are treated in a more sophisticated way. We will study these effects in future works.

Longer range interactions introduced beyond the Hartree level considered here could play a role in explaining some of the insulating states with charge modulation observed at fractional fillings and might produce some signatures at the corresponding energies. From comparison with Extended-DMFT calculations in simpler systems accounting also for intersite interactions at the DMFT level (Huang et al, PRB 90, 195114 (2014)), these features would be much weaker than the ones discussed here and they would be suppressed by nearby gates. Other interactions, like intersite exchange, much smaller than the ones include here, could play a role in stabilizing symmetry breaking states at low temperatures. The study of the states with symmetry breaking are beyond the purpose of this work.

4. "Hole (electron) doping creates the quasiparticle in the lower (upper) Hubbard band[27].

Therefore resets in the shape and position of the Hubbard bands appear around integer." Why can't fractional doping creates satellites features that has similar effects since strong correlations are important and central (analogous to fractional quantum hall effects which is integer quantum hall effects in the presence of strong correlations).

As discussed before, the local correlations (k-independent and frequency dependent self-energy) due to intersite density-density interactions could produce some weak features at the corresponding energies (notably smaller than U). These features could have some structure at fractional fillings. Our calculations, with intersite interactions treated at the Hartree level, cannot deal with these features. However, we expect the features from intersite interactions to be much weaker than the ones discussed here, and to be detected, at most, at low temperatures if a symmetry breaking does not hide them.

We do not think that the band structure we are working with would support fractional quantum hall effects (FQHE) in the absence of a magnetic field or alignment with hBN.

Reviewer #2 (Remarks to the Author):

In the submitted manuscript entitled, "Heavy quasiparticles and cascades without symmetry breaking in twisted bilayer graphene", the authors study the effects of interactions on the many-body ground state of the twisted bilayer graphene (TBG) using dynamical mean field theory (DMFT) together with Hartree corrections. The authors, in particular, focus on the "cascade" present in TBG and explain its origins within the scope of their analysis. The claimed novelty of the work lies in explaining the origins of the "cascade" without reference to symmetry breaking.

While I find the manuscript exciting and of high relevance to ongoing research in the moiré field, I have reservations regarding the authors' proposed explanation of the cascade and how it differs from the existing literature (I expand on my reasoning below).

We thank the referee for considering our manuscript exciting and of high relevance.

Until the authors clearly and convincingly answer my questions and modify the manuscript accordingly, I cannot make a recommendation for this paper to be published in a high-profile journal such as Nature Communications in its present form. If the authors can provide convincing arguments, then I am happy to recommend its publication.

Before diving into the discussion of the authors' key claim, I have the following suggestions regarding the manuscript, which, I believe, will help with the readability of the manuscript and/or authors' claims. These are:

- Could the authors provide at least a one-paragraph in-text introduction to the dynamical mean field theory and explain in what way this method is superior to Hartree-Fock, DMRG, strong-coupling calculation, or other approaches? What is its shortcoming as well? Here this point is specifically important since the authors intend to change the accepted and experimentally verified paradigm of the cascade description in the moiré systems (which

arises in all of these other calculations), the burden falls on the authors to demonstrate why and how these methods, in the authors' opinion, fail and the DMFT point of view does not.

We have included some lines in the introduction about the DMFT method and the convenience of suppressing the symmetry breaking transitions (lines 31-35) and further details in the Methods section (lines 211-215).

The applicability and suitability of each technique depends on the problem and the system under study. We give some details here about the scope and limits of each of them to argue that DMFT is specially fit to address phenomena originating in strong correlations.

Dynamical mean-field theory gives the frequency dependence of the self-energy due to the local correlations, which is the main physical ingredient of the phenomena we are discussing in this work. Many works in the last few decades have applied DMFT to reproduce the Mott transition, heavy fermion and Hund metal physics that appears in a great variety of systems including the cuprates, Fe based superconductors, heavy fermion materials, complex oxides and organic superconductors, and it is at present combined with ab-initio methods in techniques such as DFT+DMFT (Refs. [31-34]). DMFT has not been generally applied to graphene, mainly because of the energy scales of the material. In the absence of the moiré pattern, these effects are not important in graphene.

DMFT is a very expensive method in CPU and person-power. For instance, the calculation of the data in Fig. 1b and Fig. 2 has taken around 2400 days CPU time in a modern computer, (smaller number of real days thanks to the use of multiple cores).

The single site DMFT we are using in this work does not allow dealing with long range interactions (introduced at the Hartree level here) and does not account for the self-energy dependence on momentum. Such dependence would be absent in infinite dimensions. Due to the large connectivity of TBG (the triangular lattice has six nearest neighbors and hopping to far away neighbors is important) and the lack of nesting features, we expect DMFT to give even better results than in other systems, such as iron superconductors or cuprates, both with a less connected square lattice, where it has been widely applied, explaining and predicting many features observed experimentally.

The self-energy provided by Hartree-Fock depends on momentum, but not on energy; eigenstates in momentum space are infinitely long lived, missing the effects whose relevance we highlight here. For instance, Hartree-Fock does not open a gap without breaking the symmetry and therefore it cannot explain the Mott transition. When the interactions are large, the Hartree-Fock spectrum lacks accuracy.

DMRG is a computationally highly demanding technique to study correlations in 1 dimension. Application to two-dimensions is done by converting the two-dimensional system in a cylinder, with only 4-6 moiré unit cell perimeter in the case of TBG. Important improvements have been done in the last 2-3 years to include the long-range interactions, usually intractable in DMRG. Working at finite temperatures in DMRG increases considerably the computational effort. Addressing the physics that we discussed in our work requires working at temperatures above the symmetry breaking transition or being able to

suppress the symmetry breaking order, as it is otherwise difficult to identify which features are present in the normal state.

We believe that, in principle, DMRG could address the physics that we discuss but we are not sure that such calculations are feasible at present. As far as we know, so far only a very recent DMRG paper has included both the valley and spin degrees of freedom (arXiv: 2211.02693). Only one specific filling ($\nu=3$) at zero temperature was studied in that work and the remote bands, whose oscillations show up in the cascades, were not included. The focus of that paper was identifying the symmetry breaking order with the lowest energy.

Generally, strong-coupling refers to an approximation in which the kinetic energy is quenched and the electrons are assumed to be localized. Under this assumption, symmetry breaking orders have been studied by different authors. Beyond giving information on the symmetry breaking tendencies, this approximation can give a hint of some effects in the spectrum, like in the two-dot model by Yazdani in the cascades paper, but the spectrum is generally not accurate.

- I invite the authors to improve the form of the in-text citations to indicate at which point the idea was already present and discussed in the literature and at which point the view is the authors' invention. One example is the discussion in lines 18-20. Is it the authors' own idea that the physics of cascade constitutes the parent state from which the correlated phenomena emerge? If so, then I'm afraid I have to disagree, and it was introduced in Ref. 17, for example. As such, I would expect the citation to be present either after "suggesting" or at the end of the sentence in line 20. There are several more examples like that throughout the text, which the reader can misinterpret.

We thank the referee for calling our attention to this point that may lead to confusion. As the referee mentions, Ref. 17, one of papers which discovered the cascades by observing asymmetric peaks in the compressibility, already pointed out that the cascades form "the parent state out of which the more fragile superconducting and correlated insulating ground states emerge". However, in contrast to our manuscript, in that work the cascade parent state was associated to a spin/valley polarized state, as stated in the last sentence of the abstract.

Our claim goes beyond the one in Ref. [17] in the sense that we say that the cascades are a signature of a parent state which is a normal state without any symmetry breaking.

Ref. [10], the other article which discovered the cascades by looking at the reorganization of the density of states as seen by STM, also mentioned that the cascade state "characterizes the high-temperature parent phase from which various insulating and superconducting ground state phases emerge at low temperatures in magic-angle twisted bilayer graphene." That paper included a simple two-dot model where the two dots could be identified with our AAP orbitals and associated the spectral features to Hubbard bands. They also introduced a $2E_0=16$ meV splitting which effectively breaks the C_2T symmetry. Our modelling and calculations are much more advanced and we do not break any symmetry.

To avoid confusion with the messages and give appropriate credit to previous works in the manuscript we have included Refs 10 and 17 in line 20.

- The paragraph starting on line 39, in my opinion, should be expanded. Specifically, the authors should discuss their motivation or physical origin of different parameters $\epsilon = 12$, $T = 6$ K, the form of the Coulomb interaction, E.g., I imagine authors choose $T = 6$ K as it is the temperature scale at which cascade is most pronounced, but other correlated states do not yet start to manifest.

At present, there is not agreement in the literature on the best value for ϵ . As more extensively discussed in the response to the first referee we chose $\epsilon = 12$ from a comparison between a previous work by some of us (PRB 102, 155149 (2020)) and atomistic calculations by Goodwin et al (Electron. Struct. 2, 034001 (2020)) and our results are robust for larger values of ϵ (implying proportionally smaller interactions).

As for the temperature, as the referee guessed, we chose $T = 6$ K as it is a temperature scale at which the cascade is experimentally pronounced, and the low temperature correlated states do not seem to manifest. We note that we are not claiming that at 6 K all the low temperature states have already vanished.

To clarify the choice of the parameters we have included more information in the methods section (lines 201-208).

- In Fig. 1b, to help readers, I would invite the authors to annotate features on the DOS diagram.

We thank the referee for pointing us this difficulty. We have included blue and red arrows in Fig.1 and Fig.3 to help identifying the features. We have referred to these arrows in the corresponding captions and in the text (in lines 53, 61, 71, 153).

- The language throughout the manuscript needs to be sharpened and improved. Specifically, I caught a few typos, such as "cascade phenonema" in the abstract, or imprecise statements such as "first slowly and a bit faster" or "a large amount of spectral weight is visible".

Thanks for pointing these errors; we have revised the writing.

- A highly confusing part is the comparison of the Fig. 3 momentum resolved spectral weight against the Fig. 1c. For example, looking for instance at $\nu = 0.58$, assuming that $\omega = 0$ corresponds to chemical potential, the peak of the DOS at $\omega = -45$ or $\omega = 45$ meV would originate from the incoherent spectral weight. These features occur in momentum states that I would otherwise attribute to dispersive bands, c.f. Fig. 1a. Is that correct? If not, where in the states would come from dispersive bands? If indeed these are the dispersive bands, then this is in sharp disagreement with STM works (see works from Yazdani or Nadj-Perge groups) that can track the van Hove singularities of the two flat bands and their position wrt to the dispersive bands. The authors should clarify the features of these

momentum-resolved spectral plots and discuss how they connect to existing experimental works.

In Fig 3f, at $\omega=45$ or -45 meV, there is a shaded area for all momenta (it is clearer for $\omega=-45$ meV) which constitute the Hubbard bands. These incoherent states dominate the peak around these energies in the density of states plotted in Fig. 1b and Fig. 1c. This can be better seen in Extended Fig. S2, where we differentiate the contribution to Fig. 1b coming from the AAp correlated orbitals (Fig. S2a), which account for most of the spectral weight of the flat bands and are primarily responsible for the cascade features of the density of states, from the contribution of the less correlated orbitals (Fig. S2b) with more weight on the dispersive bands and little weight at low energies.

For the specific parameters chosen in this work ($\epsilon=12$, corresponding to $U=44.5$ meV, which controls the position of the Hubbard bands) and $w_0/w_1=0.78$, (which determines the energy onset of the dispersive bands with respect to the Diracs in Fig. 1a), the Hubbard and the dispersive bands overlap. This overlap would be absent for smaller w_0/w_1 and smaller U .

In hindsight, it seems that a lower value of w_0/w_1 and a larger ϵ would have given a better quantitative comparison with the experimental STM spectrum. We have checked that the cascades remain well formed for larger values of epsilon (see figure in the response to the first referee) and we believe that the qualitative features and the validity of our picture are not affected by our particular choice of parameters, which was based in previous theoretical studies.

To clarify the origin of the incoherent and coherent features, we added the arrows we mentioned before and have also modified the text in lines 59-65.

Let me now proceed to my main worry - the authors claim that the symmetry-breaking process does not explain cascade physics, and instead, it is explained by local correlations that are ONLY correctly described by the DMFT.

It was not our intention to claim in the manuscript that the local correlations are “only” correctly described by DMFT. However, DMFT is by far the technique which has been most widely used to this end over the last two decades in many different compounds. Other techniques with the ability to describe this physics are even more demanding, have sign problems, or do not allow computing certain quantities which facilitate comparison with experiment. Even the calculations that we have done in this paper would have been unthinkable a few years ago due to the computational cost.

Importantly, other authors, in the belief that the cascades were a consequence of symmetry breaking, have not looked at the effects discussed in our manuscript. Discussions on the modeling of twisted bilayer graphene even at the non-interacting level, and the widely used model in momentum space, have made it more difficult to guess the importance of correlation effects.

I recommend that authors remove or completely modify the discussion in lines 90-94, 58-59, and the abstract. Symmetry-breaking transition as described by many works on Hartree-Fock, Strong-Coupling, or DMRG calculations from Zaletel, Vishvanth, Vafeek, Berneivig, and many others (none of which the authors discuss or cite), explain or are at least consistent with the experimental observations. The authors, thus, cannot discard them and say that their DMFT description is superior. In fact, the language of symmetry breaking naturally explains recent experiments (see the Yazdani paper on the observation of IKS and TIVC states, for example), which the authors' treatment does not naturally lend itself to.

As discussed in previous answers, also to the questions of the first referee, we do not mean to say that symmetry breaking is absent in TBG, but that the cascades are already present in the normal state. Symmetry breaking will appear at lower temperatures. In this sense, Yazdani measurements, performed at 200 mK and 4 K showing IKS and TIVC, or other measurements such as those detecting ferromagnetism or charge density wave states at low temperatures are not inconsistent with our picture. In the symmetry breaking states, the reorganization of the spectral weight due to the cascades (somehow modified) will add to the effects of the order parameter making more difficult to identify the effects of the former.

To avoid confusion we have modified the text in the introduction and have reordered and included more references (21 and 23) in line 21.

Regarding citations, it is not always clear to distinguish which works deal with the cascades or just with the low temperature orders. If the referee considers that there is any specific citation that we should include and we have missed, we would be happy to know it in order to include it.

Moreover, the authors do not provide any experimental prediction of their results, just postdiction, which are not different in any experimentally measurable way from those of Hartree-Fock, Strong-Coupling, or DMRG works. As such, it becomes a matter of belief for the authors what is "correct" unless a precise, experimentally measurable difference can be identified. I would therefore recommend that authors rephrase the discussion and the whole message of the manuscript as not invalidating the symmetry-breaking transition point of view, but rather providing an alternative, but equally valid, description.

As discussed more thoroughly in the response to the first referee, the effect of strong correlations depends on the ratio between the local interaction and the kinetic energy and it should be present already in the normal state as far as the numerical technique allows to describe it.

As mentioned before, our claim is not that symmetry breaking orders predicted by other authors do not happen but that they are restricted to low temperatures. We have made changes in the text to clarify our claim: in the abstract, lines 65-68, and the conclusions.

In our work, besides describing existing experiments, we predict momentum selective incoherence (larger incoherence away from Γ in the flat bands), that we hope can be detected in further experiments (ARPES, or maybe the new Quantum Twisting Microscope). See page 6, lines 150-155 and conclusions for changes. In future work we will study the role

of strong correlations in other type of measurements which can distinguish our proposals from symmetry breaking ones.

To summarize, I enjoyed the authors' manuscript immensely. I would be more than happy to recommend its publication if the authors can make their claim about symmetry breaking more explicit. This authors can do this by either invaliding or providing clear and precise experimental predictions of how authors' description of the cascade can be differentiated from that given through the symmetry-breaking transition, which naturally emerges through the Hartree-Fock, Strong-Coupling, and the DMRG picture.

We hope that we have convincingly addressed all the referee concerns.

Reviewer #3 (Remarks to the Author):

In the manuscript under review, the authors investigate the cascade phenomenon of the parent state in twisted bilayer graphene (TBG) in terms of self-consistent DMFT+Hartree calculations. They find that the cascade phenomenon observed in spectroscopic and compressibility measurements can be explained well without spontaneous symmetry breaking, but via local moment formation and heavy-fermion physics. This is a very interesting and timely result, and the authors provide new insights that they are able to obtain due to their advanced modelling and methodology. The results are also of interest beyond TBG for strongly correlated electron systems in general. Thus, I recommend publication, in principle, if the following issues are addressed upon resubmission.

We thank the referee for considering that our work is a very interesting and timely result, suitable for publication in Nat. Comm., and that we provide new insights due to our advanced modelling and methodology.

1) The authors show that the cascade phenomenology in TBG is consistent with a strong modification of the normal state without symmetry breaking. At the same time, previous publications cited in the manuscript show that alternatively the cascade is consistent with models involving symmetry breaking transitions. Symmetry breaking is suppressed by hand in the used method. What would happen if symmetry breaking was allowed? Can the authors make clearer if their explanation is more than “just” an alternative or how to test the different scenarios experimentally.

If we allow symmetry breaking, it could occur but only at low temperatures. We expect that symmetry breaking will disappear at temperatures much smaller than the ones up to which the cascades have been detected. The picture would be: the strong correlation physics which produce the cascades survive at high temperature and corresponding signatures can be detected up to tens of K. Symmetry breaking states, if present, will originate in this distorted normal state. The nature of the symmetry breaking state will be controlled by energy scales considerably smaller than the ones involved in the cascades. We cannot say which will be the most stable symmetry breaking or whether it would be present for all integer fillings. Its nature may depend on the doping, the presence of a magnetic field or strain. In the low temperature symmetry breaking state, the incoherent features may be to

some extent modified, similar to what happens when a Mott insulator or a correlated metal close to the Mott transition orders magnetically.

In the manuscript we have emphasized that symmetry breaking can still occur at low temperatures: lines 31-33, conclusions, and methods (lines 206-207).

2) A cascade of transitions has been observed in less correlated graphene systems like Bernal bilayer and rhombohedral trilayer graphene. One interpretation is that analogous physics, but on different interaction scales, is responsible for the similarity of the observations in different graphene systems. Why do the authors think that this is not the case and TBG has to be considered differently?

As mentioned in previous answers, the importance of the strong correlations discussed here depends on the ratio between the local interaction and the kinetic energy and on the doping. The values expected for TBG, both from theory considerations and from the energy scales at which the cascade features appear experimentally, are such that strong correlations are expected to play a role, as confirmed in our manuscript. In ABC trilayer and Bernal bilayer without moiré, the energy scales and fillings refer to the carbon atoms. For the small filling of the carbon atoms in the ABC and Bernal systems, local moments are not expected to form without moiré. The transitions observed experimentally in these systems in compressibility measurements have important differences with respect to TBG in shape, doping dependence and temperature evolution and are consistent with symmetry breaking associated to specific features in the bandstructure. We note, nevertheless, that if a moiré is created, such as the one in ABC/hBN, strong correlations (with length and energy scales characteristic of the moiré unit cell) will become important. We discussed this previously for ABC/hBN (Phys. Rev. B 106, L081123 (2022)). For ABC/hBN the effect of strong correlations manifest in a somehow different form, similar to the one in Mott-like systems. The strong correlations would be prominent close to the integer fillings, and can be even responsible for the insulating behavior observed at these fillings. The role of strong correlations in ABC/hBN is consistent with recent photocurrent experiments (Science 375, 1295 (2022)).

Finally, we note that cascades very similar to the ones in TBG have been observed in STM experiments in twisted trilayer graphene (arXiv:2109.12127) for which a heavy fermion model similar to the one in TBG has been recently proposed.

To clarify the differences in the cascades in different systems, we have added a few lines to the conclusions (lines 176-183).

3) The authors use the notion of hybridization/coupling between the narrow and remote bands in TBG interchangeably with the notion of fragile topology, and thus argue that fragile topology is important for the cascade phenomenology. However, hybridization between these bands and fragile topology are not the same thing. There can be hybridization in models without fragile topology.

Since the authors do not show a comparison to a model with hybridization but without fragile topology, how can they tell which effect is the decisive one?

In TBG the fragile topology cannot be separated from the hybridization between the AAp orbitals and the lc orbitals, present around Γ of the flat bands, but there are other model systems with hybridization but without fragile topology, such as the Anderson lattice model. The behavior of the latter, which has been previously discussed in the context of heavy fermion systems, depends on the parameter regime (Jour. Phys. C 28, 455601 (2016)).

Some of the effects observed in TBG are generically linked to the hybridization, some to the presence of the lc orbitals and some of them also to the presence of the interactions between the AAp and the lc orbitals. In the new version we have clarified the role of each of these effects in the observed phenomenology (lines 132-156).

4) In a similar context, it would be very helpful if the authors could explain the relation to the heavy-fermion model of Ref. 23 which was used in Refs. 36-38 to study the cascade phenomenology in TBG. Do the results obtained from different models and methods agree? Why (not)? What would be a minimal model?

Both models are based on two strongly correlated AAp orbitals per valley and spin coupled to less correlated electrons. In our model the less correlated electrons are described by Wannier functions and the approximate particle-hole symmetry is introduced non-locally, through the fitting parameters to the continuum model. The model in Ref.23 (Ref 27 in the new version) focuses on the exact particle-hole symmetry case (they neglect a small $\sin(\theta/2)$ term in the continuum model which breaks this symmetry) and introduces this symmetry locally. To this end the less correlated electrons are introduced through a projection in k-space of the lowest states in the closest remote bands. In their model the moiré unit cell is not naturally defined, as it is in ours, they simply fold the bands. The k-dependence of the kinetic energy of the less correlated electrons and their coupling to the AAp ones is parametrized by a very simple function. This approximation is appealing as it strongly reduces the number of parameters. However, as they are obtained from a k.p expansion with respect to Γ , away from this point the range of energies at which the remote bands are well fitted is considerably limited (up to ~ 60 meV for realistic parameters).

Regarding interactions we have focused on different cases. In our work, the electrical gates are assumed to be far (we assume a $1/r$ interaction between the electrons in the carbon atoms up to a distance equal to 7 moiré unit lengths). In Ref.[27] they consider a double gated geometry, being the gates at 5 nm of the TBG so the long range interactions are strongly suppressed. It is not clear which ϵ is used.

We expect the physics from both models to be essentially the same if the same parameters (angle, dielectric constant, range of interaction) are used. So far, we are not aware of any essential difference in the results that could emerge from differences in the underlying models.

How important are the long-ranged interactions used here?

As discussed in the answer to the first referee, in our DMFT+Hartree description the long-range part of the interaction does not affect the qualitative picture discussed here. It is just giving quantitative differences in the shape of the quasiparticle band, which we will study

with more detail in future work. Going beyond our picture could produce weak features or even some states with charge modulation at fractional fillings. We have added a sentence in Methods to clarify the role of the long range interactions (line 216).

5) As a minor point: figures in Fig. 1 and Fig. 3 show many features each, so that it can be difficult to follow the discussion of these features in the text. It could be helpful to add some descriptors/arrows into the figures themselves.

We thank the referee for pointing out this difficulty. We have added some arrows to facilitate the discussion and have mentioned them in the captions and along the text (in lines 53, 61, 71, 153).

REVIEWERS' COMMENTS

Reviewer #1 (Remarks to the Author):

Having read the careful responses by the authors, I recommend that manuscript for publication, given its novelty, impact and appeal to general audience.

Reviewer #2 (Remarks to the Author):

I thank the authors for their time and effort spent in answering all of my questions. I am happy with the revisions the authors made, specifically with regard to how the subtle points of their analysis differentiate the authors' explanation of the cascade from the conventional understanding of the cascade through symmetry breaking process. The authors now further highlight how their predictions can be verified.

I recommend the publication of the manuscript in its present form.